# Optimization of cryoprotectants and storage temperatures for preserving viability and probiotic properties of lyophilized bacterial strains from chicken gut

Dipankar Sardar[1☯], Istiaq Morol[2☯], Johra Bari[3], Amalan Sarkar[4], Adnan Habib[5]*

1 Department of Biotechnology, Bangladesh Agricultural University, Mymensingh, Bangladesh,
2 Department of Microbiology and Hygiene, Bangladesh Agricultural University, Mymensingh, Bangladesh,
3 Department of Pathology and Lab Medicine, Medicare Pathology Lab, Mymensingh, Bangladesh,
4 Department of Computer Science and Engineering, Northern University of Business and Technology, Khulna, Bangladesh, 5 Animal Health Research Division, Bangladesh Livestock Research Institute, Dhaka, Bangladesh

☯ These authors contributed equally to this work.
* ahabib.bmb@gmail.com

## Abstract

Probiotics, particularly strains from the genera *Bacillus*, *Lactobacillus*, and *Staphylococcus*, play a vital role in gut health, immune modulation, and pathogen inhibition. However, environmental stressors during storage often compromise their long-term viability and probiotic functionality. By examining how lyophilization affects the viability and probiotic functionality of certain strains of *Bacillus*, *Lactobacillus*, and *Staphylococcus*, this study sought to understand how storage conditions and protective agents affect bacterial survival and important probiotic characteristics. The bacterial strains were isolated from the gastrointestinal tract of native chickens, cultivated in MRS broth, subjected to freeze-drying with different cryoprotectant formulations, and stored at varying temperatures (4°C, −20°C, and −80°C) for up to 12 months. Survival rates, stress resistance under simulated gastric and intestinal conditions, and probiotic functionality were evaluated over time. The results demonstrated that ultra-low temperature storage (−80°C) combined with a formulation of 5% glucose, 5% sucrose, 7% skim milk powder, and 2% glycine provided optimal protection. This combination effectively reduced oxidative and gastrointestinal stress and preserved key probiotic traits, including adhesion potential, antimicrobial activity, and metabolic stability. Conversely, strains stored without cryoprotectants or at higher temperatures exhibited significant viability loss and functional decline. The study highlights the critical role of optimized cryoprotection in maintaining probiotic efficacy during long-term storage. Our findings reinforce the necessity of selecting appropriate excipients and storage conditions to sustain probiotic efficacy, providing valuable insights for the development of stable, high-quality probiotic formulations. Future research should explore strain-specific responses to lyophilization and alternative preservation strategies to improve probiotic stability and performance.

**Data availability statement:** All relevant data are within the manuscript and its Supporting information files.

**Funding:** The author(s) received no specific funding for this work.

**Competing interests:** The authors have declared that no competing interests exist.

## 1. Introduction

Probiotics, defined as health-promoting beneficial microbes that confer certain benefits to their host when they are ingested in sufficient amounts, represent a rapidly expanding sector within the global functional food and pharmaceutical industries [1]. The global probiotics market, valued at approximately USD 87.70 billion in 2023, expected to grow at an annual growth rate of 14.1% to 2030, driven by increasing consumer awareness of gut health and preventive healthcare [2]. Among probiotic genera, *Bacillus*, *Lactobacillus*, and *Staphylococcus* species have gained particular attention due to their unique physiological properties and therapeutic potential. *Bacillus* species are spore-forming bacteria with exceptional stability under harsh conditions, producing diverse antimicrobial compounds including bacteriocins, lipopeptides, and enzymes [3]. *Staphylococcus* species are emerging probiotics with unique colonization abilities and immunomodulatory effects, producing specialized antimicrobial peptides and biosurfactants while demonstrating natural presence in human skin and mucosal microbiomes [4]. *Lactobacillus* species represent the most extensively studied probiotics, producing lactic acid, hydrogen peroxide, bacteriocins, and exopolysaccharides that contribute to antimicrobial activity, gut barrier protection, and enhanced nutrient absorption [5,6]. These characteristics make them valuable candidates for applications in gastrointestinal health, immune system support, and pathogen inhibition in both human and veterinary medicine.

Despite their growing global demand and therapeutic potential, maintaining the viability and functional integrity of probiotic bacteria during storage presents significant technical challenges. Environmental stressors including temperature fluctuations (>4°C variations), oxygen exposure, and moisture content above 3% can reduce bacterial viability by 2–4 log cycles within 6 months of storage [7,8]. Lactobacillus strains face significant storage challenges due to their inherent fragility as they do not have natural protective shells. The primary issue is temperature sensitivity – storage at room temperature reduces stability to just 4 weeks [9]. Bacillus and Staphylococcus strains, while generally more robust than traditional lactic acid bacteria, exhibit species-specific stress tolerances that complicate standardized preservation protocols [10]. For instance, vegetative Bacillus cells demonstrate 40–60% viability loss during conventional storage, while Staphylococcus strains can experience up to 80% reduction in colony-forming units when exposed to suboptimal conditions [8,11]. These losses not only compromise product efficacy but also result in substantial economic implications, with an estimated 20–30% of probiotic products failing to meet label claims at expiration [12]. Therefore, to effectively utilize these strains, it is essential to establish storage conditions to assure their viability and the retention of species-specific characteristics.

Lyophilization, or freeze-drying, has emerged as a preferred method for preserving probiotic strains, offering advantages such as enhanced storage stability, reduced transportation costs, and ease of handling [13]. The process involves controlled freezing at −40°C to −80°C, followed by primary drying under vacuum (0.1–0.3 mbar) and secondary drying to achieve final moisture content below 2% [14]. Despite these advantages, the lyophilization process imposes significant physiological stress on

bacterial cells including osmotic shock, protein denaturation, and membrane damage, necessitating the use of protective agents to ensure cell survival [15]. Cryoprotectants function through multiple mechanisms to preserve cellular integrity during freeze-drying. Low molecular weight carbohydrates function primarily as external osmoprotectants that maintain osmotic equilibrium across cell membranes during water removal. These compounds form hydrogen bonds with phospholipid headgroups, stabilizing membrane structure and preventing phase transitions that would otherwise compromise membrane integrity [16]. Additionally, concentrated carbohydrate solutions create vitrified matrices during freezing that physically restrict ice crystal growth and minimize mechanical cellular damage [17]. Besides, protein-based cryoprotectants form protective films around bacterial cells, buffering against rapid osmotic changes during rehydration. Furthermore, proteins help maintain the structural relationship between the cytoplasmic membrane and cell wall during the dehydration-rehydration cycle, preventing membrane detachment and subsequent cell lysis [18].

However, the complex interplay between processing parameters, protective agent composition, and bacterial stress responses during lyophilization presents several knowledge gaps specifically for Bacillus, Lactobacillus and Staphylococcus probiotic strains. While extensive research exists for lactic acid bacteria, fewer studies have systematically evaluated freeze-drying parameters for these genera. Moreover, existing studies primarily focus on cell viability metrics, with limited investigation of functional property retention including adhesion capacity, antimicrobial compound production, and immunomodulatory activity [14,19–21]. The relationship between specific cryoprotectant formulations and the maintenance of these bioactive properties remains largely unexplored. Therefore, this study aimed to optimize lyophilization protocols for Bacillus, Lactobacillus, and Staphylococcus strains isolated from chicken gut, focusing on preserving viability and probiotic functionality. Notably, the study employed a systematic evaluation of multiple cryoprotectant combinations across varying concentration gradients over an extended 12-month monitoring period—an approach that, to date, has not been thoroughly explored in probiotic preservation research. By establishing strain-specific preservation protocols, the findings will contribute to improved probiotic product development and enhanced therapeutic efficacy in commercial applications.

## 2. Materials and methods

### 2.1 Microorganisms and growth conditions

The subjects of the study were the strains *Bacillus tropicus, Bacillus tequilensis, Staphylococcus hominis, Lactobacillus salivarius,* and *Staphylococcus gallinarum* (accession number: PV082447, PV082450, PV082446, PV082449, and PV082445, respectively) obtained from Medicare Pathology Lab, Bangladesh. All strains were isolated from the gastrointestinal tract of native chickens and cultured on De Man Rogosa and Sharpe (MRS) agar (HiMedia, India) plates (pH 6.5) at 37°C for 24 hours. A single colony was then transferred to 5 mL of MRS broth and subcultured twice in the same medium. Subsequently, 1 mL of the activated culture was inoculated into 50 mL of MRS broth and incubated at 37°C for 24 hours. For long-term preservation, pure cultures in MRS broth supplemented with 15% (v/v) glycerol (D Lab Chemicals, Bangladesh) were stored at −20°C.

### 2.2 Preparation of lyophilized cells

Lyophilized cells were prepared using a lab-scale FDB-5502 freeze dryer (OPERON, South Korea) following the manufacturer's protocol with slight modifications. Bacterial strains at the mid-exponential phase of their third subculture were inoculated at 2% (v/v) into 1 L of MRS broth and incubated statically at 37°C for 24 hours. Upon reaching the early stationary phase, the cells were collected by centrifugation (10,000 × g, 10 minutes, 4°C), washed twice with sterile distilled water, and cell pellet was then concentrated five-fold to a final volume of 200 mL using fresh phosphate-buffered saline (PBS, pH 7.4) (SRL, India The concentrated suspension was divided into ten flasks (20 mL each), centrifuged once more, and then resuspended in 20 mL of various excipients—glucose, sucrose, skim milk, dextran, glycine, and glycerol—at concentrations ranging from 2% to 10% (w/v), using a 2:1 ratio of cells to excipient. Glucose and sucrose were obtained from Merck

(Germany), skim milk powder from HiMedia (India), dextran from Sigma-Aldrich (USA), glycine from SRL (India), and glycerol from D Lab Chemicals (Bangladesh). The final suspensions, each with a cell density of approximately $4 \times 10^9$ CFU/mL, were frozen at –20°C and –80°C for 18 hours, followed by lyophilization for 8 hours under a vacuum of $2 \times 10^{-2}$ Torr and a collector temperature of −50°C. After drying the tubes were plugged with rubber stoppers and covered with aluminum caps. Subsequently, the viability of the lyophilized cells was evaluated using the agar plate count method [22]. These evaluations were conducted using microorganisms rehydrated from the lyophilized powders in 1 mL of PBS at 25°C, as well as those obtained after the first subculture in MRS broth. Based on the viability results, the optimal concentration of each excipient was determined. Final lyophilization was then carried out using the selected bacterial strain ($3.8 \times 10^9$ CFU/mL) combined with all tested excipients at their optimized concentrations, following the previously described procedure. Control samples were resuspended in PBS. Details of the sample preparation and storage conditions post-lyophilization are presented in Table 1. Finally, viability, resistance to gastrointestinal and oxidative stresses, antimicrobial activity, and adhesion capacity were assessed at specific intervals over a 12-month storage period.

## 2.3 Enumeration of bacteria and percentage of viability

The plate count method [22] determined the number of viable bacteria before and after freeze-drying and during storage. Lyophilized cells were uniformly suspended in 500 µL of phosphate buffer saline (pH 7, 0.1M; 0.9% NaCl). Aliquots from the samples (0.1mL) diluted in N-saline (0.9% NaCl) (ACME, Bangladesh) were pour-plated on MRS media supplemented with 1.5% agar and incubated at 37°C for 24h. All enumerations were carried out in duplicate, and only plates with 30–300 colonies were considered for calculating the viable cell count (log CFU/mL). The number of viable cells before lyophilization ($N_0$) and after lyophilization (N) was determined, and cell viability (%) was calculated using the formula: $(N/N_0) \times 100$.

## 2.4 Determination of resistance to gastro-intestinal stress

The response of selected probiotic bacteria, reactivated after lyophilization and stored for periods ranging from 1 to 12 months, to gastric and intestinal stresses was evaluated using the Pinto method [23]. The reactivated bacterial samples were grown in a liquid medium under anaerobic conditions for 24–48 hours at 37°C and pH 6.5. To simulate the conditions of the stomach and intestines, simulated gastric fluid (SGF) (Biochemazone, India) and simulated intestinal fluid (SIF) (Biochemazone, India) were prepared. The cells in the control group received phosphate buffer treatment rather than SGF

Table 1. Variants of the preparation and storage of experimental cells using the lyophilization method.

| Condition | Active principle + Excipients* | Pre-freezing temperature | Storage temperature |
|---|---|---|---|
| I | Suspended cells in phosphate buffer saline | −20°C | 4°C |
| II | Suspended cell + Glucose + Skim milk powder+ Glycine | −20°C | −20°C |
| III | Suspended cell + Glucose + Skim milk powder + Glycine | −80°C | −20°C |
| IV | Suspended cell + Glucose + Skim milk powder + Glycine | −80°C | −80°C |
| V | Suspended cell + Sucrose + Skim milk powder + Glycine | −20°C | −20°C |
| VI | Suspended cell + Sucrose + Skim milk powder + Glycine | −80°C | −20°C |
| VII | Suspended cell + Sucrose + Skim milk powder + Glycine | −80°C | −80°C |
| VIII | Suspended cell + Glucose + Sucrose + Skim milk + Glycine | −20°C | −20°C |
| IX | Suspended cell + Glucose + Sucrose + Skim milk + Glycine | −80°C | −20°C |
| X | Suspended cell + Glucose + Sucrose + Skim milk + Glycine | −80°C | −80°C |

*Suspended cells ($3.8 \times 10^9$ CFU/mL), Glucose (5%), Sucrose (5%), Skim milk powder (7%), Glycine (2%)

and SIF. SGF is prepared by dissolving 3 g/L of pepsin (Sigma-Aldrich, USA) and 5 g/L of NaCl (Merck, Germany) in distilled water, with the pH adjusted to 2.0 using hydrochloric acid (HCl) (Merck, Germany). In the same way, SIF was made by dissolving 1 g/L of pancreatin and 10 g/L of bile salts, and the pH is adjusted to 6.8 using NaOH (Merck, Germany). Both solutions are sterilized via filtration to maintain enzyme activity. The experimental procedure was initiated by exposing the microbial culture to gastric conditions. A 1 mL sample ($3.7 \times 10^9$ CFU/mL) is mixed with 9 mL of SGF and incubated at 37°C while being continuously shaken at 100 rpm for 2 hours. Aliquots are collected at 0, 1, and 2 hours for analysis. In the case of microorganisms, serial dilutions are performed, and colony-forming units (CFU/mL) are determined using agar plating. Following gastric exposure, the surviving sample ($2.3 \times 10^6$ CFU/mL) is transferred into 9 mL of SIF to replicate the intestinal phase. The incubation continues at 37°C for an additional 4 hours with periodic sampling at 0, 2, and 4 hours. Similar analytical techniques are applied to determine microbial survival rates or bioactive compound stability over time. The resistance degree (RD) to gastric and intestinal stresses was calculated from the ratio of the number of CFU in 1 mL of the control and experimental samples. The results were evaluated based on recommendations: "very good" RD ≤ 5; "good" 5 < RD ≤ 10; "acceptable" 10 < RD ≤ 15; "unacceptable" RD > 15 [23].

## 2.5 Determination of resistance to oxygen

The relative bacterial growth ratio (RBGR) of the probiotic strains was determined with slight modifications to a standard method [24]. In brief, a probiotic culture (1 g/100 mL of broth) was incubated separately under both aerobic and anaerobic conditions at 37°C for 24 hours. After incubation, the optical density (OD) was measured at 600 nm using a spectrophotometer (ThermoFisher Scientific, USA). RBGR was measured using the following equation:

$$\text{Relative bacterial growth rate (RGBR)} = \frac{\text{OD aerobic}}{\text{OD anaerobic}}$$

## 2.6 Adhesion capabilities

The ability of lyophilized cells to adhere to the intestinal mucin layer was evaluated using a 96-well microtiter plate coated with porcine stomach mucin type II (Sigma-Aldrich, USA), following the method described by Dhanani and Bagchi (2013) [25]. A 300 µL of mucin (0.5 mg/mL) in sterile Dulbecco's PBS (Sigma-Aldrich, USA) was added to each well of the microtiter plate and incubated at 4°C overnight to allow for mucin coating. The wells were then washed twice with PBS to remove any unbound mucin. Next, 15 µL of the cell suspension was added to each well, and the final volume was adjusted to 200 µL with PBS. The plate was incubated at 37°C for 90 minutes to allow cells to adhere to the mucin layer. Afterward, unbound cells were removed by washing the wells five times with PBS. The adhered cells were then extracted using 300 µL of Triton X-100 (Sigma-Aldrich, USA) (0.05% v/v, prepared in sterile PBS) for 20 minutes at 37°C. The adherent cells were plated on MRS agar at the proper dilution to count them. After counting the colonies, the adhesion percentages were computed using the formula below:

$$\text{Adhesion index} = \frac{\text{log CFU adhered bacteria}}{\text{log CFU total bacteria}} \times 100$$

## 2.7 Antimicrobial activity

The antimicrobial effect of the selected probiotic bacteria against food spoilage and human pathogens was assessed using the spot-inoculation method with slight modifications [26]. The procedure initiated with the preparation of the test bacterial strain, typically a pathogenic bacterium (*E. coli*, *Y. enterocolitica*, *V. cholerae*, *Kl. Pneumoniae*, and *S. typhi*) which was grown in Luria-Bertani (LB) broth (HiMedia, India), until it reached the logarithmic growth phase ($OD_{600} \approx 0.5$). Next, 2 µL of the probiotic cell suspension ($2.8 \times 10^7$ CFU/mL) was applied as spots on an MRS agar plate and incubated

for 24 hours at 37°C. Then, 100 µL of the test pathogen ($3.1 \times 10^7$ CFU/mL) was mixed with 10 mL of 1% nutrient agar, which overlaid on the previously spot-inoculated MRS agar plates and incubated at 37°C for 24 hours. Following incubation, the plates were examined for zones of inhibition, which appeared as clear areas around the inoculated spots where pathogenic bacterial growth was suppressed. The zone of inhibition was measured based on recommendations: strong ≥ 20 mm, moderate < 20 mm > 10 mm, and weak ≤ 10 mm [27].

## 2.8 Statistical analysis

All experiments were conducted in triplicate and repeated at least once, with the data from representative experiments presented as the mean. The results from various experiments were analyzed using analysis of variance (ANOVA) with GraphPad Software, version 9.5.1 (GraphPad Software, Inc., USA), and mean values were compared using Tukey's range test with a significance level of $P < 0.05$.

## 3. Results

### 3.1 Selection of the concentration of excipients

The effect of different excipient concentrations (2–10%) on the viability of various bacterial strains was assessed by determining the number of viable cells (Log CFU/mL) in each condition (Fig 1). Glucose and sucrose exhibited strong cryoprotective effects at 5% and above concentrations, with no noticeable improvement at higher levels. SMP supported a steady increase in bacterial viability, with the highest viability observed at a 7% concentration. In contrast, glycine promoted the highest bacterial growth at 2%, while higher concentrations led to decreased viability. Based on these findings, different combinations of 5% glucose, 5% sucrose, 7% SMP, and 2% glycine were chosen for further experiments, as higher concentrations provided no additional benefits and were economically impractical.

### 3.2. Survival of lyophilized strains during long-term storage

The survival rates of five bacterial strains were assessed at different storage durations under ten preparation variants (Fig 2). All bacterial strains exhibited a gradual decline in viability over the storage period, with varying degrees of reduction dependent on both strain type and preparation variant. Preparation variant I consistently showed the lowest preservation efficacy for all strains, with the most dramatic reductions in viable cell counts by month 12. Notably, *S. hominis* KBC6 and *S. gallinarum* KBC10 exhibited a complete loss of viability (0 Log CFU/mL) in variant I after 12 months of storage. In contrast, preparation variants VIII, IX, and X demonstrated superior preservation properties across all strains. *L. salivarius* KBC9 maintained the highest viability throughout the storage period, retaining 5.15–5.74 Log CFU/mL after 12 months depending on the preparation variant. The most substantial improvement in preservation was observed between variants I and II, with subsequent variants (III-X) showing more incremental enhancements in maintaining bacterial viability. Among the preparation variants, Variant X consistently exhibited the highest survival rates, followed by Variants VIII and IX, across most strains and storage periods, suggesting that the choice of preparation method significantly influences bacterial viability during storage.

### 3.3 Determination of resistance to gastrointestinal stress

The study evaluated the gastric stress resistance of five probiotic bacterial strains over five storage periods (0, 3, 6, 9, and 12 months) across ten distinct preparation variants (Table 2). *B. tropicus* KBC8 maintained "very good" resistance for 3 months but gradually declined to "acceptable" levels, with variants VII–X showing better stability. *B. tequilensis* KBC4 exhibited "very good" resistance for 3 months, with a gradual decline; by 12 months, variants VI–X retained "good" resistance while I–V became "acceptable." *L. salivarius* KBC9 showed strong resistance for 6 months, with variants VI–X maintaining "good" stability after 12 months. *S. hominis* KBC6 displayed "very good" resistance early on, but

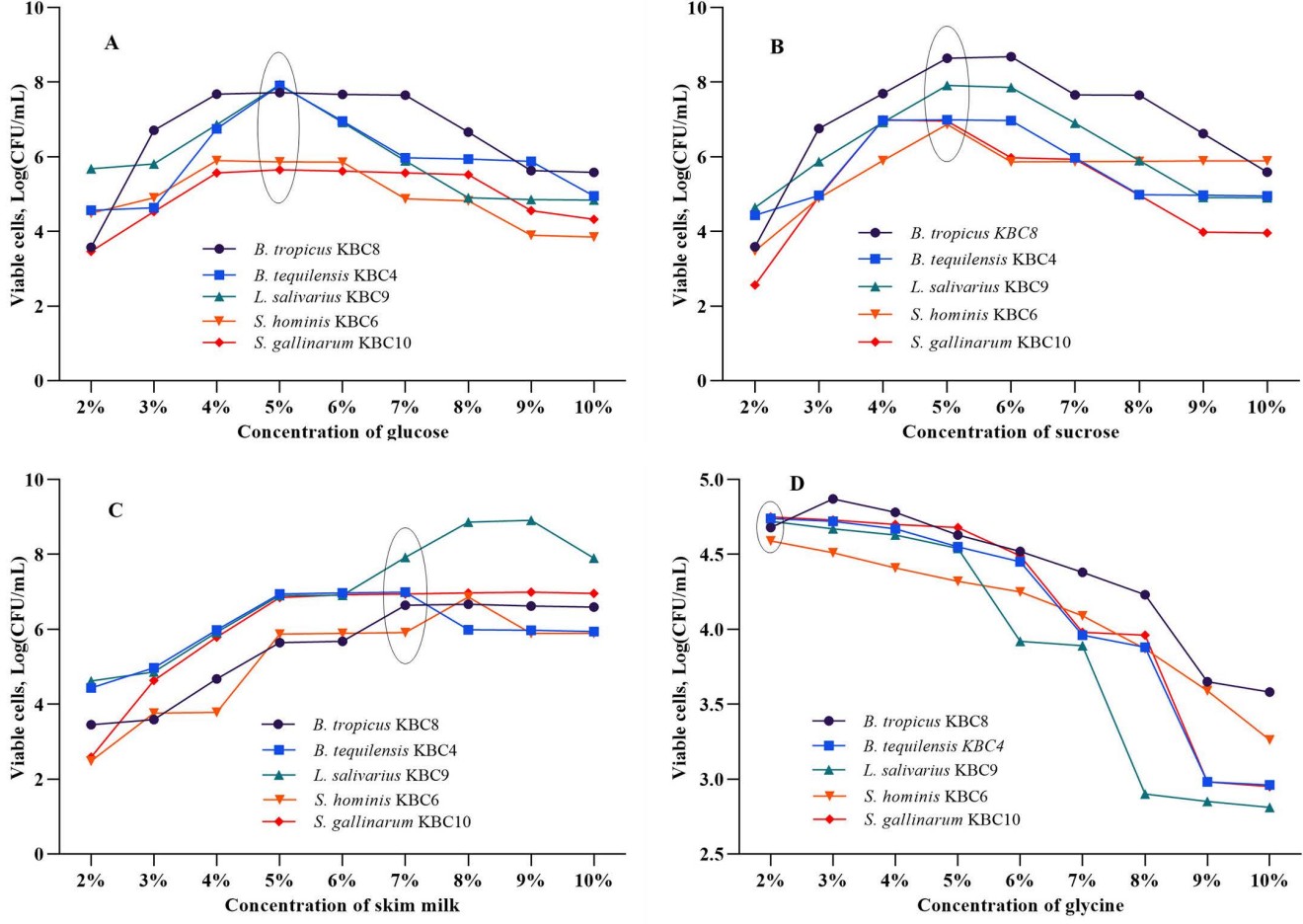

**Fig 1. Effect of different excipient concentrations on the viability of probiotic strains.** Viable cell counts (log CFU/mL) of five probiotic strains—*Bacillus tropicus* KBC8, *Bacillus tequilensis* KBC4, *Lactobacillus salivarius* KBC9, *Staphylococcus hominis* KBC6, and *Staphylococcus gallinarum* KBC10—were measured after exposure to varying concentrations (2–10%) of four excipients: (A) glucose, (B) sucrose, (C) skim milk, and (D) glycine. Each panel represents the effect of one excipient on the viability of all five strains. The experiments were conducted in triplicate, and the data are expressed as mean values. Ellipses indicate the concentration at which the maximum viability was observed for each excipient across the tested strains, representing the optimal concentration for formulation.

by 12 months, variants I–II approached the "unacceptable" threshold, whereas VII–X maintained "good" resistance. *S. gallinarum* KBC10 showed an initial decline in resistance by 6 months, with variants VII–X demonstrating better long-term stability.

The study also assessed the intestinal stress resistance of the selected bacterial strains over five storage periods (0, 3, 6, 9, and 12 months) across ten different preparation variants (Table 3). Over 12 months of storage, intestinal stress resistance declined across all probiotic strains, with significant variation among preparation variants. Variants VII–X consistently demonstrated better long-term stability, while variants I–V showed greater resistance loss, often reaching "acceptable" or near-"unacceptable" levels. All strains initially exhibited very good resistance (RD ≤ 5), but by 9–12 months, resistance dropped, particularly in variants I–V. The findings emphasize the impact of preparation methods on probiotic stability under intestinal stress conditions.

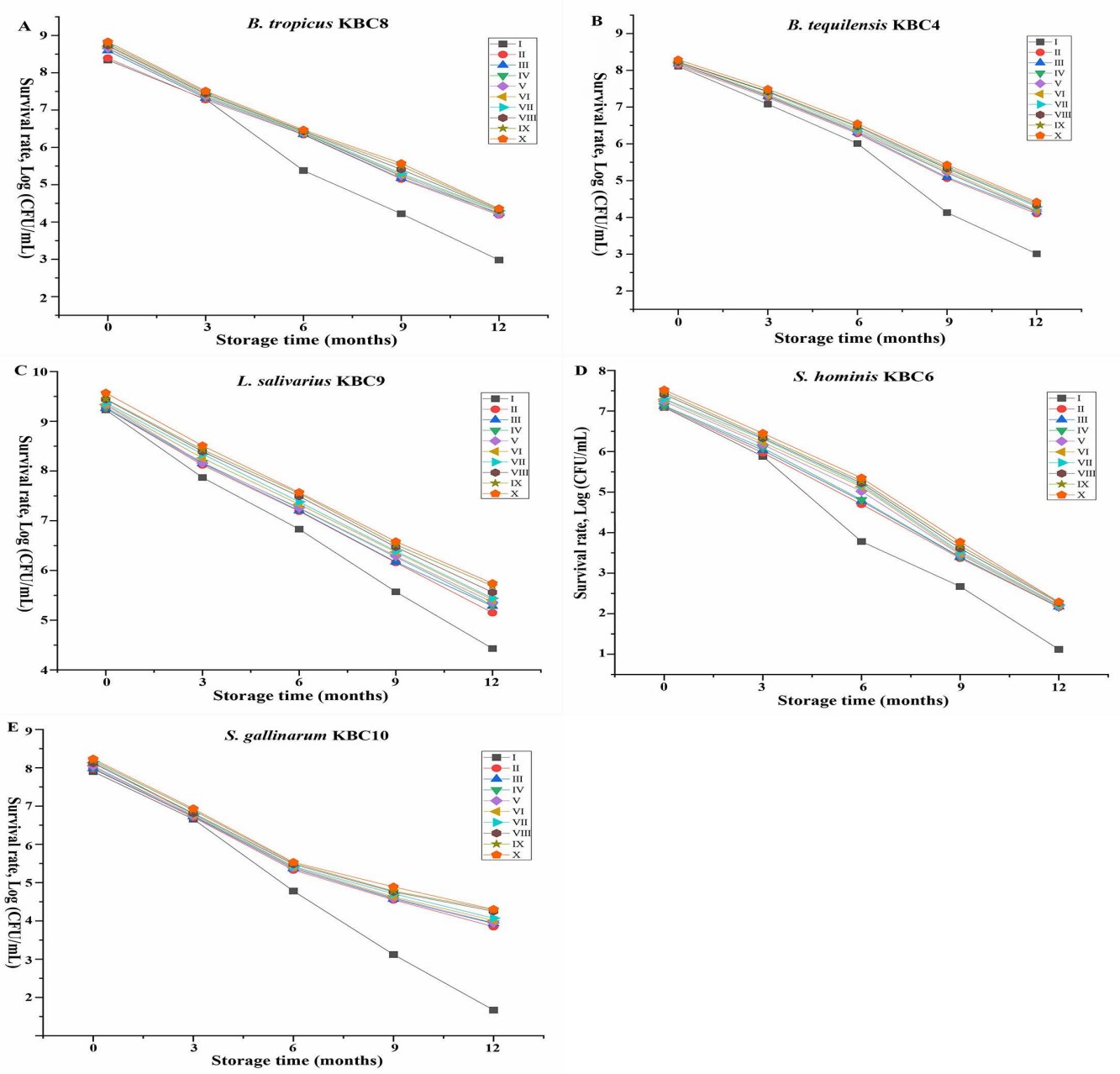

**Fig 2. Long-term survival of lyophilized probiotic bacterial strains under different formulation conditions.** Survival rates (log CFU/mL) of lyophilized (A) *Bacillus tropicus* KBC8, (B) *Bacillus tequilensis* KBC4, (C) *Lactobacillus salivarius* KBC9, (D) *Staphylococcus hominis* KBC6, and (E) *Staphylococcus gallinarum* KBC10 were assessed over 12 months of storage. Bacterial preparations were stored under ten different formulation conditions (variants I–X) as indicated in the legends. All experiments were performed in triplicate, and the data are presented as mean.

## 3.4 Determination of resistance to oxidative stress

Over the 12-month storage period, the relative bacterial growth ratio (RBGR) of all five strains showed a clear decline, indicating a reduction in bacterial viability (Table 4). Initially, at month 0, RBGR values were relatively high (ranging from 2.0 to 2.5), suggesting optimal conditions for bacterial growth. However, by month 3 and month 6, the RBGR dropped

Table 2. Resistance of selected probiotic isolates to gastric stress at different stages of storage under different preparation variants.

| Strain | Storage time (months) | Variants for the preparation | | | | | | | | | | SEM | P value |
|---|---|---|---|---|---|---|---|---|---|---|---|---|---|
| | | RD score (Resistance to gastric stress, exposure time 30 min) | | | | | | | | | | | |
| | | I | II | III | IV | V | VI | VII | VIII | IX | X | | |
| *B. tropicus* KBC8 | 0 | 3.00a | 2.00a | 2.00a | 1.00a | 2.00a | 1.00a | 1.00a | 1.00a | 1.00a | 1.00a | 0.260 | 0.07016 |
| | 3 | 5.00b | 4.00b | 3.00a | 2.00a | 3.00a | 3.00a | 2.00a | 2.00a | 2.00a | 2.00a | 0.327 | 0.00029 |
| | 6 | 7.00c | 6.00c | 6.00c | 5.00c | 6.00c | 4.00a | 4.00a | 3.00b | 2.00b | 2.00d | 0.563 | 0.00020 |
| | 9 | 10.00d | 7.00c | 7.00c | 6.00c | 6.00c | 6.00c | 6.00c | 5.00a | 5.00a | 4.00b | 0.512 | 0.00457 |
| | 12 | 13.00a | 10.00a | 10.00a | 10.00a | 10.00a | 9.00a | 8.00a | 9.00a | 8.00a | 7.00a | 0.521 | 2.81323 |
| *B. tequilensis* KBC4 | 0 | 2.00a | 2.00a | 2.00a | 1.00a | 2.00a | 2.00a | 1.00a | 2.00a | 2.00a | 1.00a | 0.153 | 0.09032 |
| | 3 | 3.00a | 3.00a | 3.00a | 3.00a | 3.00a | 3.00a | 2.00a | 3.00a | 2.00a | 2.00a | 0.153 | 0.08017 |
| | 6 | 8.00b | 6.00b | 6.00b | 5.00c | 6.00b | 6.00b | 4.00c | 5.00b | 3.00a | 3.00a | 0.489 | 0.00001 |
| | 9 | 10.00d | 9.00b | 9.00b | 7.00c | 8.00b | 7.00c | 6.00c | 7.00c | 6.00a | 6.00a | 0.453 | 0.04451 |
| | 12 | 11.00a | 11.00a | 10.00a | 10.00a | 10.00a | 8.00b | 7.00c | 7.00c | 6.00c | 6.00c | 0.636 | 0.05726 |
| *L. salivarius* KBC9 | 0 | 2.00a | 2.00a | 1.00a | 1.00a | 2.00a | 1.00a | 1.00a | 1.00a | 1.00a | 1.00a | 0.153 | 0.07013 |
| | 3 | 4.00a | 3.00a | 3.00a | 2.00b | 3.00a | 2.00a | 2.00b | 2.00b | 2.00b | 2.00b | 0.223 | 0.00062 |
| | 6 | 5.00b | 5.00b | 5.00b | 5.00b | 4.00c | 4.00c | 3.00a | 3.00a | 3.00a | 3.00a | 0.298 | 0.00571 |
| | 9 | 9.00c | 7.00c | 7.00c | 6.00c | 7.00c | 6.00c | 6.00c | 5.00a | 4.00b | 4.00b | 0.482 | 0.04781 |
| | 12 | 12.00b | 11.00b | 10.00b | 9.00b | 10.00b | 9.00b | 8.00b | 8.00b | 7.00a | 7.00a | 0.526 | 0.05281 |
| *S. hominis* KBC6 | 0 | 4.00b | 3.00a | 2.00a | 1.00c | 3.00a | 2.00a | 1.00c | 2.00a | 2.00a | 1.00c | 0.314 | 0.02357 |
| | 3 | 5.00b | 5.00b | 4.00b | 4.00b | 5.00b | 3.00a | 2.00a | 3.00a | 2.00c | 1.00c | 0.452 | 0.05321 |
| | 6 | 8.00a | 8.00a | 7.00a | 7.00a | 7.00a | 6.00c | 6.00c | 5.00b | 5.00b | 3.00b | 0.489 | 0.00824 |
| | 9 | 13.00c | 12.00c | 11.00c | 10.00c | 11.00c | 9.00a | 9.00a | 10.00a | 8.00a | 7.00a | 0.577 | 0.03675 |
| | 12 | 17.00a | 17.00a | 14.00a | 13.00a | 13.00a | 13.00a | 11.00a | 10.00a | 9.00a | 9.00b | 0.921 | 0.02548 |
| *S. gallinarum* KBC10 | 0 | 5.00a | 4.00a | 4.00a | 3.00a | 4.00a | 4.00a | 3.00a | 4.00a | 2.00b | 2.00b | 0.307 | 0.00124 |
| | 3 | 7.00c | 6.00c | 6.00c | 6.00c | 6.00c | 5.00a | 5.00a | 5.00a | 4.00b | 3.00d | 0.367 | 0.00021 |
| | 6 | 11.00d | 10.00d | 9.00d | 9.00d | 10.00d | 8.00b | 8.00b | 7.00c | 4.00a | 3.00a | 0.823 | 0.03211 |
| | 9 | 14.00c | 11.00c | 11.00c | 11.00c | 11.00c | 10.00c | 9.00a | 9.00a | 8.00b | 7.00b | 0.623 | 0.00678 |
| | 12 | 16.00b | 15.00b | 14.00b | 14.00b | 13.00b | 13.00b | 11.00a | 11.00a | 10.00a | 9.00c | 0.718 | 0.05421 |

All experiments were performed in triplicate and the experimental data are presented as mean. SEM- Standard error mean, RD- resistance degree; "very good"- RD ≤ 5; "good"- 5 < RD ≤ 10; "acceptable"- 10 < RD ≤ 15; "unacceptable"- RD > 15.

[a,b,c,d]Means in each row with different superscripts are significantly (*P* < 0.05) different from each.

moderately to around 1.5–2.0, reflecting a decrease in bacterial survival. By month 9, the decline was more pronounced, with RBGR values around 1.3–1.7, showing an increase in stress on the bacteria. By month 12, the lowest RBGR values were observed, approximately 1.0–1.5, indicating significant viability loss across the strains. However, variants VII–X showed better growth retention than variants I–V, maintaining higher RBGR values, suggesting that specific preparation methods may offer better protection during prolonged storage.

## 3.5 Adhesion capabilities

The adhesion index of all five bacterial strains exhibited a gradual decline over the 12-month storage period, indicating reduced bacterial adhesion potential as the storage duration increased (Fig 3). For *B. tropicus* KBC8, the adhesion index started at a range of 21–29% at month 0, with a slight decrease to 9–16% by month 12. Similarly, *B. tequilensis* KBC4 showed a decrease from an initial range of 20–26% to 10–17% by month 12. Both strains demonstrated the most considerable decline in adhesion in the later months, with month 12 showing the lowest values. In contrast, *L. salivarius* KBC9, a strain with a relatively higher adhesion index at the start (ranging from 57% to 65% at month 0), showed a more moderate

**Table 3. Resistance of selected probiotic isolates to intestinal stress at different stages of storage under different preparation variants.**

| Strain | Storage time (months) | Variants for the preparation | | | | | | | | | | SEM | P value |
|---|---|---|---|---|---|---|---|---|---|---|---|---|---|
| | | RD score (Resistance to intestinal stress, exposure time 5 h) | | | | | | | | | | | |
| | | I | II | III | IV | V | VI | VII | VIII | IX | X | | |
| *B. tropicus* KBC8 | 0 | 3.00[a] | 3.00[a] | 3.00[a] | 2.00[a] | 3.00[a] | 2.00[a] | 1.00[a] | 2.00[a] | 1.00[a] | 1.00[a] | 0.276 | 0.08445 |
| | 3 | 3.00[b] | 3.00[b] | 2.00[b] | 2.00[b] | 2.00[a] | 2.00[a] | 1.00[c] | 2.00[a] | 1.00[c] | 1.00[c] | 0.233 | 0.00075 |
| | 6 | 5.00[b] | 5.00[b] | 5.00[b] | 4.00[b] | 5.00[b] | 4.00[c] | 4.00[c] | 3.00[c] | 3.00[a] | 2.00[a] | 0.333 | 0.00004 |
| | 9 | 10.00[a] | 6.00[b] | 6.00[b] | 6.00[b] | 6.00[b] | 5.00[b] | 5.00[b] | 4.00[b] | 3.00[c] | 3.00[c] | 0.635 | 0.04534 |
| | 12 | 16.00[d] | 11.00[b] | 11.00[b] | 10.00[b] | 11.00[b] | 10.00[b] | 9.00[c] | 8.00[c] | 7.00[c] | 6.00[a] | 0.874 | 0.02478 |
| *B. tequilensis* KBC4 | 0 | 5.00[a] | 4.00[a] | 4.00[a] | 4.00[a] | 4.00[a] | 3.00[a] | 3.00[a] | 2.00[b] | 2.00[b] | 2.00[b] | 0.334 | 0.00811 |
| | 3 | 5.00[b] | 4.00[b] | 4.00[b] | 3.00[a] | 4.00[b] | 3.00[a] | 3.00[a] | 3.00[a] | 2.00[c] | 1.00[c] | 0.359 | 0.00463 |
| | 6 | 7.00[b] | 7.00[b] | 7.00[b] | 6.00[b] | 7.00[b] | 6.00[a] | 5.00[d] | 5.00[d] | 3.00[a] | 2.00[a] | 0.562 | 0.00581 |
| | 9 | 11.00[a] | 10.00[a] | 10.00[a] | 9.00[a] | 10.00[a] | 9.00[a] | 9.00[a] | 8.00[b] | 7.00[b] | 7.00[b] | 0.421 | 0.01667 |
| | 12 | 15.00[c] | 12.00[c] | 11.00[c] | 10.00[c] | 11.00[c] | 10.00[b] | 10.00[b] | 8.00[a] | 7.00[a] | 6.00[d] | 0.816 | 0.05223 |
| *L. salivarius* KBC9 | 0 | 2.00[a] | 2.00[a] | 2.00[a] | 1.00[a] | 2.00[a] | 1.00[a] | 1.00[a] | 1.00[a] | 1.00[a] | 1.00[a] | 0.163 | 0.50057 |
| | 3 | 3.00[b] | 3.00[b] | 3.00[b] | 2.00[a] | 3.00[b] | 2.00[a] | 2.00[a] | 2.00[a] | 2.00[a] | 1.00[c] | 0.213 | 0.00004 |
| | 6 | 5.00[a] | 5.00[a] | 5.00[a] | 4.00[b] | 5.00[a] | 5.00[a] | 4.00[b] | 3.00[c] | 2.00[c] | 2.00[c] | 0.394 | 0.00449 |
| | 9 | 10.00[b] | 7.00[b] | 7.00[b] | 6.00[b] | 7.00[b] | 6.00[c] | 5.00[a] | 5.00[a] | 4.00[a] | 4.00[d] | 0.567 | 0.03721 |
| | 12 | 16.00[a] | 12.00[a] | 11.00[a] | 11.00[a] | 10.00[b] | 10.00[b] | 8.00[d] | 8.00[d] | 7.00[c] | 6.00[c] | 0.912 | 0.70121 |
| *S. hominis* KBC6 | 0 | 4.00[a] | 4.00[a] | 4.00[a] | 2.00[c] | 4.00[a] | 3.00[a] | 1.00[b] | 2.00[c] | 1.00[b] | 1.00[b] | 0.426 | 0.04789 |
| | 3 | 8.00[b] | 8.00[b] | 7.00[b] | 7.00[b] | 6.00[b] | 5.00[c] | 5.00[c] | 6.00[b] | 4.00[c] | 2.00[a] | 0.592 | 0.00075 |
| | 6 | 10.00[b] | 9.00[a] | 7.00[a] | 7.00[a] | 8.00[a] | 7.00[a] | 6.00[c] | 6.00[c] | 6.00[c] | 5.00[d] | 0.482 | 0.00967 |
| | 9 | 15.00[a] | 13.00[a] | 13.00[a] | 12.00[a] | 12.00[b] | 12.00[b] | 10.00[c] | 11.00[b] | 9.00[c] | 8.00[c] | 0.654 | 0.00759 |
| | 12 | 17.00[b] | 16.00[b] | 15.00[b] | 15.00[b] | 13.00[a] | 13.00[a] | 12.00[a] | 12.00[a] | 10.00[c] | 9.00[c] | 0.814 | 0.00063 |
| *S. gallinarum* KBC10 | 0 | 3.00[a] | 3.00[a] | 2.00[a] | 2.00[a] | 2.00[a] | 1.00[a] | 1.00[a] | 2.00[a] | 1.00[a] | 1.00[a] | 0.249 | 0.50757 |
| | 3 | 7.00[a] | 6.00[b] | 5.00[b] | 5.00[b] | 6.00[b] | 5.00[b] | 4.00[c] | 4.00[c] | 3.00[c] | 2.00[d] | 0.473 | 0.00005 |
| | 6 | 11.00[b] | 10.00[b] | 9.00[c] | 9.00[c] | 8.00[c] | 8.00[c] | 7.00[c] | 7.00[c] | 6.00[a] | 4.00[a] | 0.640 | 0.01923 |
| | 9 | 15.00[c] | 15.00[c] | 13.00[c] | 12.00[b] | 12.00[b] | 11.00[b] | 11.00[b] | 10.00[a] | 8.00[a] | 7.00[d] | 0.832 | 0.03444 |
| | 12 | 18.00[a] | 17.00[a] | 15.00[a] | 14.00[b] | 14.00[b] | 13.00[b] | 11.00[d] | 11.00[d] | 10.00[c] | 10.00[c] | 0.895 | 0.00523 |

All experiments were performed in triplicate and the experimental data are presented as mean. SEM- Standard error mean, RD- resistance degree; "very good"- RD ≤ 5; "good"- 5 < RD ≤ 10; "acceptable"- 10 < RD ≤ 15; "unacceptable"- RD > 15.

[a,b,c,d]Means in each row with different superscripts are significantly ($P < 0.05$) different from each.

decrease over time, from 13–41% at month 12. While the adhesion index dropped, it maintained a higher level compared to the other strains throughout the storage period, indicating a stronger ability to adhere even after prolonged storage. For *S. hominis* KBC6 and *S. gallinarum* KBC10, the adhesion indices also decreased over time. *S. hominis* KBC6 showed a decline from 24–31% at month 0–5–19% by month 12. *S. gallinarum* KBC10 started with 30–38% adhesion at month 0 and dropped to 7–21% by month 12.

## 3.6 Antimicrobial activity

The antimicrobial activity of the selected probiotic bacteria against five pathogenic bacteria (*E. coli*, *Y. enterocolitica*, *V. cholerae*, *Kl. pneumoniae*, and *S. typhi*) was evaluated across ten different lyophilization preparation variants over a 12-month storage period (S1 data). For *B. tropicus* KBC8, all pathogens were sensitive for 6 months. By 12 months, *V. cholerae* exhibited the broadest resistance (variants I, II, III, V, VI, and VIII), while variants VII, IX, and X maintained full antimicrobial efficacy. *B. tequilensis* KBC4 showed a similar trend, with variants I lose effectiveness against all pathogens by 12 months, whereas variants III, VI, VII, IX, and X remained stable. *L. salivarius* KBC9 exhibited stable antimicrobial

**Table 4. Relative bacterial growth ratio (RBGR) values of the selected probiotic strains at different stages of storage under different preparation variants.**

| Strain | Storage time (months) | Variants for the preparation | | | | | | | | | | SEM | P value |
| --- | --- | --- | --- | --- | --- | --- | --- | --- | --- | --- | --- | --- | --- |
| | | Relative bacterial growth ratio (RBGR) | | | | | | | | | | | |
| | | I | II | III | IV | V | VI | VII | VIII | IX | X | | |
| B. tropicus KBC8 | 0 | 2.01a | 2.03a | 2.04a | 2.07a | 2.08a | 2.08a | 2.11a | 2.13a | 2.16a | 2.17a | 0.017 | 0.70021 |
| | 3 | 1.68b | 1.92a | 1.92a | 1.93a | 1.95a | 1.97c | 1.98c | 2.02d | 2.04d | 2.07d | 0.033 | 0.00018 |
| | 6 | 1.42b | 1.76a | 1.78a | 1.79a | 1.81c | 1.81c | 1.83c | 1.85c | 1.86d | 1.89d | 0.041 | 0.00621 |
| | 9 | 1.13a | 1.62b | 1.63b | 1.65b | 1.66b | 1.68b | 1.70d | 1.73b | 1.75c | 1.76c | 0.057 | 0.00456 |
| | 12 | 1.02c | 1.31a | 1.33a | 1.33a | 1.35a | 1.36a | 1.38c | 1.42d | 1.43d | 1.47d | 0.039 | 0.05129 |
| B. tequilensis KBC4 | 0 | 2.31a | 2.34a | 2.37a | 2.39a | 2.41a | 2.44a | 2.49c | 2.50c | 2.53b | 2.54b | 0.025 | 0.00202 |
| | 3 | 1.87b | 1.92a | 1.95c | 1.99c | 2.04c | 2.08a | 2.09a | 2.14a | 2.15d | 2.18d | 0.033 | 0.04178 |
| | 6 | 1.66a | 1.81b | 1.82b | 1.83b | 1.85b | 1.87c | 1.89c | 1.93d | 1.95d | 1.99d | 0.029 | 0.00811 |
| | 9 | 1.43a | 1.52c | 1.52c | 1.54c | 1.61c | 1.63c | 1.65b | 1.68b | 1.68b | 1.69d | 0.027 | 0.00041 |
| | 12 | 1.27c | 1.33b | 1.34b | 1.36b | 1.39b | 1.40a | 1.43a | 1.47d | 1.49d | 1.53d | 0.025 | 0.02153 |
| L. salivarius KBC9 | 0 | 2.24b | 2.27b | 2.26a | 2.28a | 2.31c | 2.32c | 2.33c | 2.36d | 2.39d | 2.43d | 0.019 | 0.00017 |
| | 3 | 2.06a | 2.12c | 2.15c | 2.17c | 2.21d | 2.22d | 2.26d | 2.29b | 2.31b | 2.32b | 0.027 | 0.00485 |
| | 6 | 1.69b | 1.76a | 1.77a | 1.81a | 1.83a | 1.85a | 1.89c | 1.90c | 1.93c | 1.95c | 0.026 | 0.00081 |
| | 9 | 1.32a | 1.36a | 1.36 | 1.37b | 1.38a | 1.40a | 1.43a | 1.47d | 1.48d | 1.51c | 0.019 | 0.05081 |
| | 12 | 1.06b | 1.17a | 1.18a | 1.19a | 1.21a | 1.22a | 1.25a | 1.29c | 1.30c | 1.30c | 0.023 | 0.04777 |
| S. hominis KBC6 | 0 | 2.22c | 2.23c | 2.25c | 2.25c | 2.28a | 2.29a | 2.33a | 2.38b | 2.39b | 2.42b | 0.022 | 0.00498 |
| | 3 | 2.01a | 2.05a | 2.06a | 2.08b | 2.13c | 2.14c | 2.15c | 2.21d | 2.23d | 2.24d | 0.025 | 0.00011 |
| | 6 | 1.77b | 1.83a | 1.83a | 1.84a | 1.84a | 1.86a | 1.86a | 1.87a | 1.88c | 1.90c | 0.011 | 0.04382 |
| | 9 | 1.53a | 1.57a | 1.59a | 1.59a | 1.60c | 1.62c | 1.65d | 1.67d | 1.67d | 1.69b | 0.038 | 0.00818 |
| | 12 | 1.27b | 1.31c | 1.32c | 1.32c | 1.35c | 1.36a | 1.39a | 1.38a | 1.40d | 1.41d | 0.025 | 0.14055 |
| S. gallinarum KBC10 | 0 | 2.34b | 2.37a | 2.38a | 2.41a | 2.43a | 2.43a | 2.46c | 2.51d | 2.52d | 2.57d | 0.023 | 0.05022 |
| | 3 | 2.26a | 2.29a | 2.30a | 2.30a | 2.34b | 2.35b | 2.36b | 2.39b | 2.38c | 2.40c | 0.014 | 0.00087 |
| | 6 | 1.78a | 2.07b | 2.07b | 2.09b | 2.12c | 2.12c | 2.15c | 2.17c | 2.18c | 2.21d | 0.038 | 0.00353 |
| | 9 | 1.37b | 1.88c | 1.89c | 1.88c | 1.88c | 1.89c | 1.91a | 1.93a | 1.92a | 1.95d | 0.053 | 0.00261 |
| | 12 | 1.12a | 1.47c | 1.48b | 1.51b | 1.50b | 1.51b | 1.51b | 1.53c | 1.55d | 1.58d | 0.040 | 0.06582 |

All experiments were performed in triplicate, and the experimental data are presented as mean.

a,b,c,dMeans in each row with different superscripts (a, b, c, and d) are significantly ($P<0.05$) different from each.

activity, with *E. coli* remaining sensitive throughout. By 12 months, *Kl. pneumoniae* showed the highest resistance (variants I, II, III, V, VI, and VIII), while variants IV, VII, IX, and X demonstrated superior stability. *S. hominis* KBC6 showed increasing resistance, particularly in *V. cholerae*, which was resistant to all variants except X for 12 months. Variants IV, VII, IX, and X exhibited the best stability. For *S. gallinarum* KBC10, E. coli remained sensitive for 6 months but developed resistance in several variants by 12 months. *Y. enterocolitica* and *V. cholerae* showed complete resistance at 12 months, while *Kl. pneumoniae* and *S. typhi* displayed variant-specific resistance patterns. Overall, variants VII, IX, and X provided the most stable antimicrobial activity across all strains, supporting their use for long-term probiotic preservation.

## 4. Discussion

The optimization of probiotic preservation represents a critical challenge in biotechnology, requiring careful balance between maintaining cellular viability and preserving functional properties during long-term storage. This study provides comprehensive insights into the complex interplay between cryoprotectant formulation, storage conditions, and bacterial strain characteristics in determining preservation success.

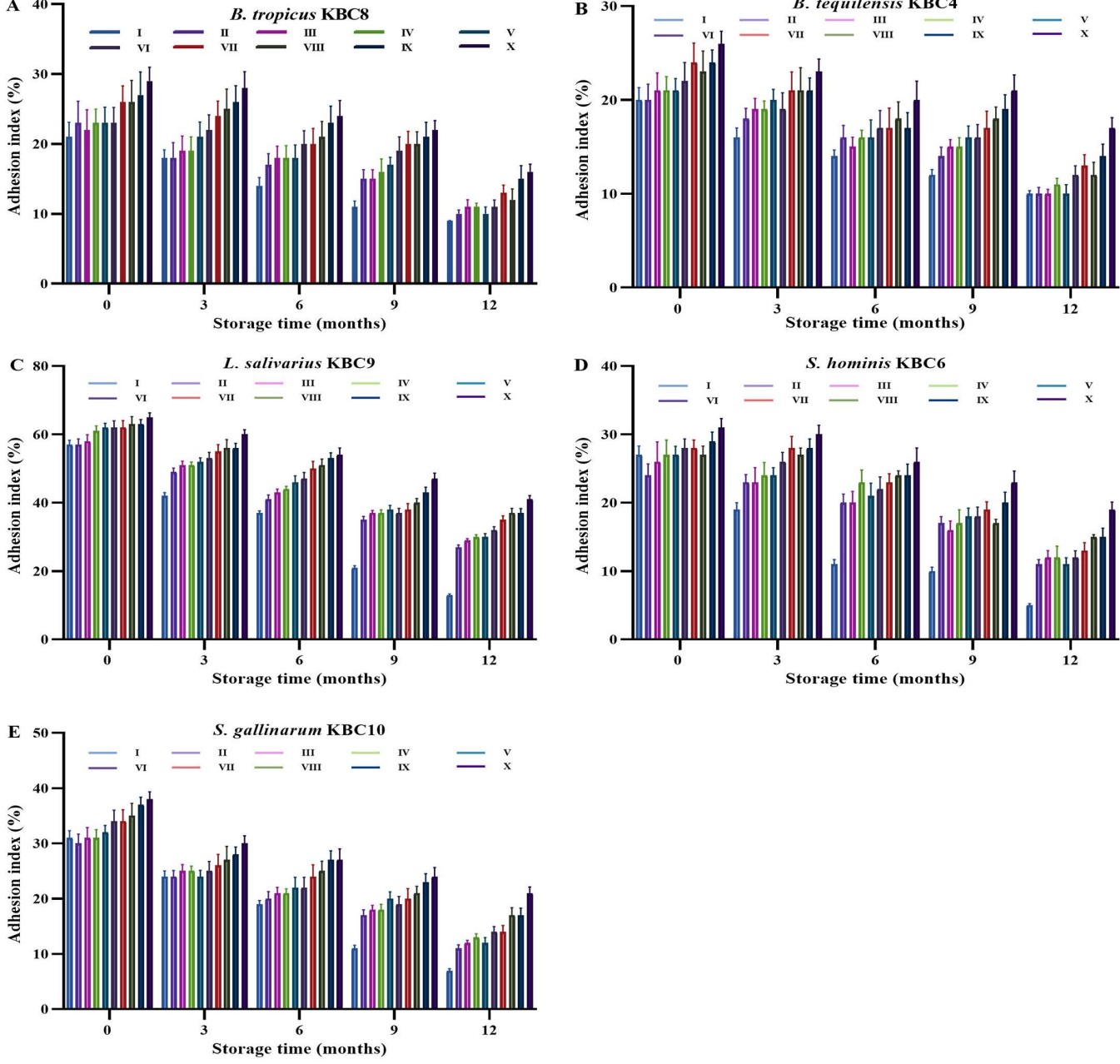

**Fig 3. Adhesion ability of lyophilized probiotic bacterial strains during long-term storage under different formulation conditions.** Adhesion index (%) of lyophilized (A) *Bacillus tropicus* KBC8, (B) *Bacillus tequilensis* KBC4, (C) *Lactobacillus salivarius* KBC9, (D) *Staphylococcus hominis* KBC6, and (E) *Staphylococcus gallinarum* KBC10 measured at 0, 3, 6, 9, and 12 months of storage under ten different preparation variants (I–X). Adhesion ability was evaluated as the percentage of bacterial cells adhering to intestinal epithelial cells. Each bar represents the mean of triplicate experiments; error bars indicate standard deviations.

Our findings reveal distinct concentration-dependent responses among different cryoprotective agents, highlighting the importance of precise formulation optimization. The optimal protective concentrations identified—5% for glucose and sucrose, 7% for skim milk powder, and 2% for glycine—reflect the unique mechanisms by which each agent confers

cellular protection. The cryoprotective efficacy of sugars (5% glucose or sucrose) aligns with established research showing they stabilize cell membranes during freeze-drying by replacing water molecules and forming hydrogen bonds with membrane phospholipids [28]. Beyond this threshold, additional sugar molecules provide diminishing returns, suggesting saturation of available membrane binding sites. SMP demonstrated superior protection at 7% concentration due to its complex composition. The casein proteins in SMP provide amphipathic interactions with bacterial membranes while simultaneously creating protective protein matrices during freezing [29]. The additional benefits of reduced water activity, pH buffering, and nutrient supplementation create a comprehensive protective environment that extends beyond simple membrane stabilization [30]. The gradual increase in protection up to 7% suggests a dose-dependent interaction between these multiple mechanisms that maximize at this concentration, beyond which the protective effect plateaus are due to saturation of binding sites or excessive viscosity limiting heat and mass transfer during processing [31]. The biphasic response observed with glycine presents both opportunities and challenges for formulation development. While 2% glycine provides effective protein stabilization through compatible solute mechanisms [32], higher concentrations become counterproductive, potentially disrupting osmotic balance and intracellular pH regulation [33]. The synergistic multi-component approach combining all four agents represents rational formulation design, addressing membrane integrity, protein stability, and osmotic regulation simultaneously. This comprehensive strategy provides redundant protection mechanisms, ensuring preservation efficacy even when individual components may be partially compromised during processing or storage.

The temperature-dependent preservation results underscore the critical importance of storage conditions in maintaining probiotic viability. The complete viability loss observed in simple PBS storage at 4°C, particularly for Staphylococcus strains, demonstrates the inadequacy of basic preservation approaches for sensitive bacterial species [34]. Ultra-low temperature storage at −80°C consistently outperformed intermediate freezing temperatures, confirming that reduced molecular mobility and minimized ice crystal formation are essential for long-term preservation [35]. The incremental improvements observed across different temperature conditions suggest that preservation optimization requires consideration of both pre-freezing treatment and long-term storage parameters. The exceptional resilience of Lactobacillus strain across all storage conditions provides valuable insights into strain selection for probiotic applications. This stability likely reflects evolutionary adaptations that enhance stress tolerance, including robust membrane compositions and efficient stress response systems [36]. Conversely, the poor performance of Staphylococcus strains highlights critical gaps in stress response mechanisms among certain staphylococcal species. Their inability to maintain functionality even with optimal cryoprotection suggests fundamental limitations in membrane stability or stress gene regulation that may limit their commercial viability without genetic or physiological enhancement. Understanding these intrinsic characteristics could inform both strain selection strategies and targeted preservation approaches for different bacterial species.

The preservation of gastrointestinal stress resistance emerged as a critical indicator of probiotic functionality retention. The temporal decline observed across all strains reflects the progressive deterioration of cellular stress response mechanisms during extended storage [37]. However, the strain-specific patterns reveal important differences in stress tolerance mechanisms. Lactobacillus starins' superior gastric acid resistance maintenance suggests robust acid tolerance systems, likely including efficient proton pump mechanisms and pH homeostasis pathways [38]. The enhanced bile tolerance observed in Bacillus strains compared to Staphylococcus species indicates differential bile salt hydrolase activity and redundant efflux system efficiency [39]. These findings suggest that stress resistance mechanisms may serve as predictive markers for overall preservation success. The poor stress resistance preservation in basic storage conditions (Variant I) emphasizes the interconnected nature of cellular preservation and functional maintenance. Membrane damage from inadequate cryoprotection compromises not only viability but also the membrane-associated systems essential for stress response [40].

The progressive decline in relative bacterial growth rate provides insights into the hierarchical nature of storage-induced cellular damage. Early storage periods showed moderate impacts on growth potential, suggesting initial damage

of membrane phospholipids and sulfhydryl oxidation of key metabolic enzymes, impairing nutrient transport and energy production [41,42]. Extended storage revealed more severe compromise of fundamental cellular processes, including DNA replication and protein synthesis machinery [43]. The superior RBGR preservation in optimized formulations (Variants VII-X) demonstrates that comprehensive cryoprotection can maintain cellular machinery integrity. The proposed mechanisms—DNA stabilization by sugars, ribosomal protection by milk proteins, and enzyme stabilization by glycine—provide a framework for understanding multi-component preservation strategies [44–46]. The correlation between cryoprotectant complexity and RBGR maintenance suggests that metabolic activity preservation requires protection of multiple cellular systems simultaneously. This finding has implications for probiotic product development, where maintained growth potential is essential for therapeutic efficacy.

Bacterial adhesion capacity preservation represents a critical functional parameter for probiotic applications, particularly those requiring intestinal colonization. The differential adhesion retention among strains highlights the importance of surface protein stability and exopolysaccharide production during storage [47]. The temperature-dependent adhesion preservation confirms that surface structures are particularly vulnerable to freeze-thaw damage. The superior protection provided by ultra-low temperature storage likely results from reduced ice crystal formation and minimized mechanical disruption of cell surface components [48]. The enhanced adhesion preservation in multi-component formulations suggests that surface protein stabilization requires comprehensive protection strategies. The combination of membrane stabilization, protein matrix formation, and pH buffering provides multiple protective mechanisms for maintaining surface functionality [49].

The decline in antimicrobial activity during storage represents a significant concern for probiotic therapeutic applications. The mechanisms underlying this decline—reduced bacteriocin production, compromised secretion systems, and decreased organic acid production—reflect the complex nature of antimicrobial function preservation [50,51]. Lactobacillus strains' superior antimicrobial activity retention correlates with its overall stress resistance and growth maintenance, suggesting that antimicrobial function preservation depends on comprehensive cellular integrity. The strain's ability to maintain salivaricin production indicates robust transcriptional and secretory systems [52]. The pathogen-specific responses observed, particularly V. cholerae's rapid resistance development, highlight the dynamic nature of antimicrobial interactions during storage. Understanding these pathogen-specific responses could inform targeted preservation strategies for specific therapeutic applications [53].

The parallel decline observed in multiple functional parameters—adhesion, antimicrobial activity, and stress resistance—suggests shared mechanistic bases for functional property preservation. This correlation indicates that comprehensive preservation strategies targeting fundamental cellular integrity will simultaneously protect multiple therapeutic characteristics. The optimal preservation combination identified in this study—multi-component cryoprotectant formulation with ultra-low temperature storage—provides a practical framework for commercial probiotic development. However, the strain-specific responses observed emphasize the need for tailored preservation protocols based on individual bacterial characteristics.

## 5. Conclusion

This study highlights the critical role of lyophilization and optimized cryoprotective formulations in maintaining the viability and probiotic functionality of selected Bacillus, Lactobacillus and Staphylococcus strains. Using sugars (glucose, sucrose), skim milk powder, and glycine as protective agents significantly enhanced bacterial survival, stress resistance, and long-term storage stability, particularly under ultra-low temperatures (−80°C). Key probiotic traits, including adhesion potential, antimicrobial activity, and metabolic stability, declined over time but were best preserved in optimized formulations. Our findings reinforce the necessity of selecting appropriate excipients and storage conditions to sustain probiotic efficacy, providing valuable insights for the development of stable, high-quality probiotic formulations. Future research should explore strain-specific responses to lyophilization and alternative preservation strategies to further improve probiotic stability and performance.

## Supporting information

**S1 Dataset. Antimicrobial activity_*B. tropicus* KBC8, antimicrobial activity_*B. tequilensis* KBC4, antimicrobial activity_*L. salivarius* KBC9, antimicrobial activity_*S. hominis* KBC6, antimicrobial activity_*S. gallinarum* KBC10.** (XLSX)

## Author contributions

**Conceptualization:** Dipankar Sardar, Adnan Habib.

**Data curation:** Dipankar Sardar, Istiaq Morol, Johra Bari.

**Formal analysis:** Dipankar Sardar, Amalan Sarkar.

**Investigation:** Dipankar Sardar, Istiaq Morol, Adnan Habib.

**Methodology:** Dipankar Sardar, Istiaq Morol, Adnan Habib.

**Resources:** Dipankar Sardar, Johra Bari, Amalan Sarkar.

**Supervision:** Adnan Habib.

**Validation:** Dipankar Sardar, Istiaq Morol, Adnan Habib.

**Visualization:** Dipankar Sardar, Istiaq Morol, Adnan Habib.

**Writing – original draft:** Dipankar Sardar.

**Writing – review & editing:** Dipankar Sardar, Istiaq Morol, Johra Bari, Amalan Sarkar, Adnan Habib.

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
