## [Decision Letter · Decision Letter 0]

PONE-D-25-19776Optimization of lyophilization process for long-term storage of probiotic Bacillus and Staphylococcus: Effects on survival and functionalityPLOS ONE

Dear Dr. Habib,

Thank you for submitting your manuscript to PLOS ONE. After careful consideration, we feel that it has merit but does not fully meet PLOS ONE’s publication criteria as it currently stands. Therefore, we invite you to submit a revised version of the manuscript that addresses the points raised during the review process.

We look forward to receiving your revised manuscript.

Kind regards,

Mohammad Faezi Ghasemi, Ph.D

Academic Editor

PLOS ONE

Journal Requirements:

Additional Editor Comments :

Dear respected Authors,

Your manuscript entitled" Optimization of lyophilization process for long-term storage of probiotic Bacillus and Staphylococcus: Effects on survival and functionality was evaluated and based on our reviewers

suggestions needs revisions. So, please follows the reviewers comments and prepare point by point responses for all details. Due to use different Bacillus Spp. , staphylococcus Spp. and one strain of Lactobacillus salivarius, indicate in the title in a correct way.

Best regards,

M.Faezi (PhD)

Reviewers' comments:

Reviewer's Responses to Questions

**Comments to the Author**

1. Is the manuscript technically sound, and do the data support the conclusions?

Reviewer #1: Partly

Reviewer #2: Yes

2. Has the statistical analysis been performed appropriately and rigorously? 

Reviewer #1: Yes

Reviewer #2: Yes

3. Have the authors made all data underlying the findings in their manuscript fully available?

Reviewer #1: Yes

Reviewer #2: Yes

4. Is the manuscript presented in an intelligible fashion and written in standard English?

Reviewer #1: Yes

Reviewer #2: Yes

5. Review Comments to the Author

Reviewer #1: The data presented here are interesting. However, there are several shortcomings reduce the value of the manuscript. The language definitely requires attention, there are several typos and grammatical errors. Additionally, some parts are poorly described or confused (see also below).

Introduction:

-The manuscript requires improvement in the Introduction.

- What is the novelty of your study?

Material and Methods:

- Please add references for all protocols.

- Add name and reference of the mediums and materials.

- Antimicrobial activity section misses important data and assays may not have been conducted as recommended by antibiotic susceptibility testing guidelines. According to Clinical and Laboratory Standard Institute (CLSI) and European committee for antibiotic susceptibility testing (EUCAST) the determination of MICs with the method must use bacterial inoculum standardized at 5 x 105 CFU/ml that does not seem to be the case here. Did you control the number of CFU/ml in your inoculum? Which OD cut-off was considered positive growth? If the antimicrobial testing was not performed according to guidelines, it must be discussed in the manuscript.

- What is test bacterial strain??

Results

- The figures 1, 2, and 3 are very illegible, the quality of the charts should be improved.

- I strongly advise authors to revise the explanation of the figures. Legends of many figures has to be increased.

- What does RD mean in the tables legends?

Discussion

- Some of the content presented here are repetition of the results.

- I feel that the Discussion is too long and can be shorted by 25% or so easily without losing important information.

-Some of the content presented here are subject to the Introduction.

- I also would recommend a better organization of ideas in the Discussion.

Reviewer #2: - Strengthen the introduction by highlighting the probiotic species of Bacillus and Staphylococcus.

-Line 111, Bacterial cells washed twice with sterile distilled water. Won't washing bacteria with distilled water lead to their osmotic lysis?

-In Table one, mention the used concentration of each compound in preparation of storage conditions.

-In Table1 (Suspended cells), what are the washed cells suspended in?

-In2.4 A 1 mL sample is mixed with 9 mL of SGF. Mention the CFU of bacteria in 1 mL of sample.

-According to 2.4, the text mentions that following gastric exposure, the surviving sample is transferred into 9 mL of SIF to replicate the intestinal phase. As the population of bacteria is important in their resistance rate, in related results, mention the population of survived bacteria for second test.

- Please include reference for each section of methods.

-The effect of different excipient concentrations (2–10%) on the viability of various bacterial strains was not assessed in methods. The related results are presented in 3.1.

-In Line 202, how do you find Skim Milk Powder is the most effective excipient for probiotic strains?

-For 3.6, authors need to substantiate pathogenic bacterial growth inhibition with plate photographs.

6. PLOS authors have the option to publish the peer review history of their article (what does this mean? ). If published, this will include your full peer review and any attached files.

**Do you want your identity to be public for this peer review?** For information about this choice, including consent withdrawal, please see our Privacy Policy .

Reviewer #1: No

Reviewer #2: No

---

## [Author Response · Author response to Decision Letter 1]

12 Jun 2025

Additional Editor Comments:

Due to use different Bacillus Spp., staphylococcus Spp. and one strain of Lactobacillus salivarius, indicate in the title in a correct way.

Response: Optimization of cryoprotectant formulations and storage temperatures for preserving viability and probiotic properties of lyophilized bacterial strains from chicken gut (L1-3)

Reviewer #1:

Introduction:

Comment: The manuscript requires improvement in the Introduction.

Response: We have thoroughly revised this section to enhance its clarity, logical flow, and relevance to the study. In particular, we have enhanced the background by including recent literature to better highlight the importance of the research topic, clearly identified the knowledge gap that our study aims to address and provided a more focused rationale for the study objectives.

Comment: What is the novelty of your study?

Response: The study employed a systematic evaluation of multiple cryoprotectant combinations across varying concentration gradients over an extended 12-month monitoring period—an approach that, to date, has not been thoroughly explored in probiotic preservation research.

Material and Methods:

Comment: Please add references for all protocols.

Response: All protocols used in this study were supported by appropriate references to ensure methodological validity and reproducibility (L125, L140, L155, L166, L188, L196, L209).

Comment: Add name and reference of the mediums and materials.

Response: The names and references of all media and materials used in this study have been clearly provided (L119, L123, L131, L134-136, L158, L168-173, L195, L197, L202, L212).

Comment: Antimicrobial activity section misses important data and assays may not have been conducted as recommended by antibiotic susceptibility test guidelines. According to the Clinical and Laboratory Standard Institute (CLSI) and European committee for antibiotic susceptibility testing (EUCAST) the determination of MICs with the method must use bacterial inoculum standardized at 5 x 105 CFU/ml that does not seem to be the case here. Did you control the number of CFU/ml in your inoculum? Which OD cut-off was considered positive growth? If the antimicrobial testing was not performed according to guidelines, it must be discussed in the manuscript.

Response: Antimicrobial activity was evaluated using a modified version of the spot-inoculation method. The concentrations of both the test pathogens and probiotic inoculum were specified in terms of CFU/mL (L213-214). In this study, we considered OD600> 0.1 as indicative of positive growth.

Comment: What is test bacterial strain??

Response: The test bacterial strains were typically pathogenic bacteria. In this study, five pathogenic strains—Escherichia coli, Yersinia enterocolitica, Vibrio cholerae, Klebsiella pneumoniae, and Salmonella typhi—were used to evaluate the antimicrobial activity of the probiotic candidates.

Results

Comment: The figures 1, 2, and 3 are very illegible, the quality of the charts should be improved.

Response: We have enhanced the resolution and clarity of these figures to ensure they are fully legible. We have also adjusted the font sizes, color contrast, and line thickness where necessary to improve visual quality and readability.

Comment: I strongly advise authors to revise the explanation of the figures. Legends of many figures has to be increased.

Response: We have carefully revised the figure legends throughout the manuscript to provide more detailed and comprehensive explanations.

Comment: What does RD mean in the table’s legends?

Response: RD means resistance degree. It refers to what extent an organism's ability to withstand or counteract a particular factor, such as a disease, a chemical, or an environmental stressor.

Discussion

Comment:

-Some of the content presented here is repetition of the results.

-I feel that the Discussion is too long and can be shorted by 25% or so easily without losing important information.

-Some of the content presented here is subject to the Introduction.

-I also would recommend a better organization of ideas in the Discussion.

Response: We have revised the Discussion section accordingly. Redundant content overlapping with the Results and Introduction has been removed, and the length has been reduced as much as possible without compromising key information. We have also reorganized the section to improve clarity and logical flow of ideas. We believe these changes have strengthened the overall quality of the manuscript.

Reviewer #2:

Comment: Strengthen the introduction by highlighting the probiotic species of Bacillus and Staphylococcus.

Response: We have revised the Introduction by emphasizing the relevance of probiotic species within the genera Bacillus and Staphylococcus. Specifically, we now highlight their roles in promoting gut health, enhancing immune response, and improving nutrient absorption (L51-63).

Comment: Line 111, Bacterial cells washed twice with sterile distilled water. Won't washing bacteria with distilled water lead to their osmotic lysis?

Response: Yes, washing bacterial cells with sterile distilled water can lead to osmotic lysis. However, in this study, the risk of lysis was minimized by limiting exposure time and handling the cells gently as per manufacturer's protocol. Moreover, all the bacterial strains examined were Gram-positive, which are inherently more resistant due to their thick peptidoglycan cell walls.

Comment: In Table one, mention the used concentration of each compound in preparation of storage conditions.

Response: The concentration of each compound is provided in the table footnote (L152-153).

Comment: In Table1 (Suspended cells), what are the washed cells suspended in?

Response: The washed cells suspended in Phosphate Buffered Saline.

Comment: In 2.4 A 1 mL sample is mixed with 9 mL of SGF. Mention the CFU of bacteria in 1 mL of sample.

Response: CFU/mL has been included (L175).

Comment: According to 2.4, the text mentions that following gastric exposure, the surviving sample is transferred into 9 mL of SIF to replicate the intestinal phase. As the population of bacteria is important in their resistance rate, in related results, mention the population of survived bacteria for second test.

Response: CFU/mL has been included (L179).

Comment: Please include reference for each section of methods.

Response: All protocols used in this study were supported by appropriate references to ensure methodological validity and reproducibility (L125, L140, L155, L166, L188, L196, L209).

Comment: The effect of different excipient concentrations (2–10%) on the viability of various bacterial strains was not assessed in methods. The related results are presented in 3.1.

Response: Thank you for pointing this out. We acknowledge the omission and have now revised the Methods section to include a detailed description of how the effects of different excipient concentrations (2–10%) on the viability of the bacterial strains were assessed (Section 2.2).

Comment: In Line 202, how do you find Skim Milk Powder the most effective excipient for probiotic strains?

Response: We have revised the Results section to clearly state that Skim Milk Powder showed a steady improvement in bacterial viability, with optimal effects observed at a concentration of 7%.

Comment: For 3.6, authors need to substantiate pathogenic bacterial growth inhibition with plate photographs.

Response: Thank you for your valuable comment. We conducted the antimicrobial test over a 12-month period, evaluating 10 variants in triplicate during each trial. Given the extensive number of plates generated, including all images in the main manuscript was impractical. Therefore, we have summarized the data using inhibition zone ranges and provided detailed results in five supplementary tables.

---

## [Decision Letter · Decision Letter 1]

Optimization of cryoprotectants and storage temperatures for preserving viability and probiotic properties of lyophilized bacterial strains from chicken gut

PONE-D-25-19776R1

Dear Dr. Habib,

We’re pleased to inform you that your manuscript has been judged scientifically suitable for publication and will be formally accepted for publication once it meets all outstanding technical requirements.

Kind regards,

Mohammad Faezi Ghasemi, Ph.D

Academic Editor

PLOS ONE

Additional Editor Comments (optional):

Reviewers' comments:

Reviewer's Responses to Questions

**Comments to the Author**

1. If the authors have adequately addressed your comments raised in a previous round of review and you feel that this manuscript is now acceptable for publication, you may indicate that here to bypass the “Comments to the Author” section, enter your conflict of interest statement in the “Confidential to Editor” section, and submit your "Accept" recommendation.

Reviewer #1: (No Response)

Reviewer #2: All comments have been addressed

2. Is the manuscript technically sound, and do the data support the conclusions?

Reviewer #1: (No Response)

Reviewer #2: Yes

3. Has the statistical analysis been performed appropriately and rigorously? 

Reviewer #1: (No Response)

Reviewer #2: Yes

4. Have the authors made all data underlying the findings in their manuscript fully available?

Reviewer #1: (No Response)

Reviewer #2: Yes

5. Is the manuscript presented in an intelligible fashion and written in standard English?

Reviewer #1: (No Response)

Reviewer #2: Yes

6. Review Comments to the Author

Reviewer #1: (No Response)

Reviewer #2: (No Response)

7. PLOS authors have the option to publish the peer review history of their article (what does this mean? ). If published, this will include your full peer review and any attached files.

**Do you want your identity to be public for this peer review?** For information about this choice, including consent withdrawal, please see our Privacy Policy .

Reviewer #1: No

Reviewer #2: No

---

## [Editor Report · Acceptance letter]

PONE-D-25-19776R1

PLOS ONE

Dear Dr. Habib,

I'm pleased to inform you that your manuscript has been deemed suitable for publication in PLOS ONE. Congratulations! Your manuscript is now being handed over to our production team.

Kind regards,

on behalf of

Dr. Mohammad Faezi Ghasemi

Academic Editor

PLOS ONE